

# Efficient virus-induced gene silencing in *Hibiscus hamabo* Sieb. et Zucc. using tobacco rattle virus

Zhiquan Wang[1], Xiaoyang Xu[1], Longjie Ni[1,2], Jinbo Guo[1] and Chunsun Gu[1]

[1] Jiangsu Key Laboratory for the Research and Utilization of Plant Resources, Institute of Botany, Jiangsu Province and Chinese Academy of Sciences, Nanjing, Jiangsu, China
[2] College of Forest Sciences, Nanjing Forestry University, Nanjing, Jiangsu, China

## ABSTRACT

**Background**. *Hibiscus hamabo* Sieb. et Zucc. is a semi-mangrove plant used for the ecological restoration of saline-alkali land, coastal afforestation and urban landscaping. The genetic transformation *H. hamabo* is currently inefficient and laborious, restricting gene functional studies on this species. In plants, virus-induced gene silencing provides a pathway to rapidly and effectively create targeted gene knockouts for gene functional studies.

**Methods**. In this study, we tested the efficiency of a tobacco rattle virus vector in silencing the cloroplastos alterados 1 (*CLA1*) gene through agroinfiltration.

**Results**. The leaves of *H. hamabo* showed white streaks typical of *CLA1* gene silencing three weeks after agroinfiltration. In agroinfiltrated *H. hamabo* plants, the *CLA1* expression levels in leaves with white streaks were all significantly lower than those in leaves from mock-infected and control plants.

**Conclusions**. The system presented here can efficiently silence genes in *H. hamabo* and may be a powerful tool for large-scale reverse-genetic analyses of gene functions in *H. hamabo*.

Corresponding author
Chunsun Gu, chunsungu@cnbg.net, chunsungu@126.com

## INTRODUCTION

*Hibiscus hamabo* Sieb. et Zucc., which is a shrub plant in the genus *Hibiscus*, family Malvaceae, is an important semi-mangrove plant (*Nakanishi, 1979*). Because of its excellent salt tolerance and morphological characteristics, *H. hamabo* is widely used in public parks, waysides and coastal sands near sea level (*Fowler, 2017*; *Li et al., 2012*; *Yang, Du & Wang, 2008*). In addition, *H. hamabo* is a good plant material for exploring the salt-stress response mechanisms of woody plants (*Li et al., 2012*). Gene manipulation technologies can be used to determine the gene functions and regulatory mechanisms in *H. hamabo*. However, to date, the inefficient and laborious genetic transformation procedures used have impeded such research. Additionally, transcriptome analyses have mined many excellent genes that are awaiting functional identification. Appropriate techniques need to be applied successfully to allow the study of gene functions in this plant.

Virus-induced gene silencing (VIGS) is a powerful technology that uses engineered viruses to specifically silence host gene expression through post-transcriptional gene silencing (*Becker & Lange, 2010*; *Krishnan et al., 2015*; *Purkayastha & Dasgupta, 2009*). VIGS is an effective method for large-scale analysis of genes and their functions, and it has been successfully performed in many plants, including tobacco, *Arabidopsis*, tomato, cotton, wheat, and many woody plants (*Burch-Smith et al., 2006*; *Jiang et al., 2014*; *Kumagai et al., 1995*; *Orzaez et al., 2009*; *Scofield et al., 2005*). VIGS works via a mechanism that is similar to that of RNA interference (*Baulcombe, 1999*; *Baulcombe, 2004*; *Burch-Smith et al., 2004*; *Lu et al., 2003*; *Waterhouse, Wang & Lough, 2001*). Double-stranded (ds) RNA is the key to the VIGS process; the dsRNA can be cleaved into short interfering (si) RNAs of 21 to 25 nucleotides (*Burch-Smith et al., 2004*; *Jiang et al., 2014*; *Lu et al., 2003*). Two strands can be obtained from the siRNAs: the guide and passenger strands. The RNA-induced silencing complex incorporates the guide strand to degrade the specific single-stranded RNA that is complementary to the guide RNA, and then, the passenger strand is degraded (*Mustafa et al., 2016*). As a result, the target gene is silenced and large amounts of siRNAs are produced (*Fuchs, Damm-Welk & Borkhardt, 2004*).

Agrobacterium-mediated VIGS protocols based on tobacco rattle virus (TRV) have been developed and optimized in cotton, and previous studies showed that TRV is a useful vector for VIGS in *Gossypium* species (*Gao et al., 2011*; *Ge et al., 2016*). Tobacco rattle virus (TRV), belonging to genus *Tobravirus* (family Virgaviridae), is a suitable virus vector system for VIGS (*Jiang et al., 2014*). A positive sense single-stranded RNA genome exists in TRV, consisting of two components, RNA 1 and RNA 2 (*Mustafa et al., 2016*). RNA 1 encodes genes with viral replication and movement functions, while RNA 2 encodes the coat protein and some nonessential structural proteins that can be replaced by foreign sequences (*Hayward, Padmanabhan & Dinesh-Kumar, 2011*). The TRV vector has been used in *G.* spp., *Arabidopsis* and *Vernicia fordii* to silence the cloroplastos alterados 1 (*CLA1*) gene, which is involved in chloroplast development (*Jiang et al., 2014*; *Manhães, De Oliveira & Shan, 2015*; *Mustafa et al., 2016*). The *CLA1* gene is highly conserved in various plant species (*Jiang et al., 2014*). The silencing phenotypes of albino leaves were observed in *Vernicia fordii* two weeks after inoculation using a heterologous TRV-based VIGS system, in which *CLA1* was isolated from *Populus tomentosa* Carr. (*Jiang et al., 2014*). The silenced *CLA1* is a useful marker for determining silencing efficiency because of the bleached phenotype (*Mustafa et al., 2016*).

In this study, we tested the feasibility of the TRV-VIGS system in *H. hamabo* using the *HhCLA1* gene as a reporter. The agroinfiltrated leaves of *H. hamabo* showed white streaks typical at three weeks after infection, and the expression levels of the *HhCLA1* gene in leaves with white streaks were significantly lower than those in leaves from mock-infected and control plants. Thus, the TRV-VIGS system can efficiently silence genes in *H. hamabo*. To our knowledge, this is the first report of the successful application of VIGS in *H. hamabo*.

**Table 1** Primers used in this TRV-VIGS system.

| Primer name | Primer sequence |
|---|---|
| *HhCLA1*-F | CTGTGAGTAAGGTTACCGAATTCTCATGTTGTCACTGAGAAAGG |
| *HhCLA1*-R | CTCGAGACGCGTGAGCTCCATAGCAAATCTTACAGGCAG |
| *qHhCLA1*-F | CGCCAGGGAACAAAGGGGTT |
| *qHhCLA1*-R | AATCGTGCATCCGCGACAGT |
| *18S rRNA*-F | GGTCGGATTTGGAACGGCGA |
| *18S rRNA*-R | CTCCACGGGCGTATCGAGG |

**Notes.**
Underlines indicate restriction enzyme cleavage sites used in this TRV-VIGS system.

# MATERIALS & METHODS

## Plant materials and growth conditions

Seeds of *H. hamabo* were collected from Nanjing's Sun Yat-Sen Memorial Botanical Garden. The seeds were then treated with concentrated sulfuric acid for 15 min and rinsed thoroughly with sterile water. The pretreated seeds were sown into flowerpots containing a mixture of peat and vermiculite (1: 1, v: v) in an illuminated incubator with controlled temperatures of 26 °C/22 °C under a 16 h/8 h (day/night) photoperiod.

## Sequence analysis

Based on the *HhCLA1* sequence (GenBank accession no. MK229167), the deduced protein sequence was analyzed with CLA1 proteins of other species using ClustalX (*Liu et al., 2015*). The amino acid sequences were obtained from NCBI (https://www.ncbi.nlm.nih.gov/). Then, the sequences were used to construct a phylogenetic tree, which was drawn with MEGA 7.0 using the Neighbor-Joining (NJ) method and 1,000 bootstrap replicates.

## VIGS vector construction

Total RNA was extracted from the leaves of *H. hamabo* using a Plant RNeasy Mini Kit (Qiagen, Hilden, Germany). The first-strand cDNA was synthesized using a SuperScript II reverse transcriptase kit (TaKaRa, Dalian, China). The primer pair *HhCLA1*-F and *HhCLA1*-R (Table 1) was designed using Oligo 6.0 software (Molecular Biology Insights, Inc., Cascade, CO, USA) based on the conserved domain of *HhCLA1*. To amplify partial fragments of *HhCLA1*, the primer pair, cDNA and PrimeSTAR™ HS DNA polymerase (TaKaRa) were used. *Eco*RI enzyme cleavage sites were added to the upstream primers and *Sac* I enzyme cleavage sites were added to the downstream primers. PCR product were generated with the following reaction program: 30 cycles of 98 °C for 10 s, 60 °C for 5 s and 72 °C for 1 min. The reactions final volume was 50 μL, containing 25 μL of 2× PrimeSTAR™ GC Buffer, 4 μL dNTP mixture (2.5 mM), 0.2 μM of each primer (final), 100 ng of cDNA and 0.5 μL of PrimeSTAR™ HS DNA Polymerase (2.5 U/μL). The *pTRV1* and *pTRV2* vectors were used in this study as described previously (*Gao et al., 2011*; *Liu, Schiff & Dinesh-Kumar, 2002*). The PCR products were ligated into pTRV2 (Fig. S1) (double-digested with *Eco*RI and *Sac* I enzymes) using a ClonExpress® IIOne Step Cloning Kit (Vazyme, Nanjing, China). The resulting vector was designated pTRV2-*HhCLA1*.

## Agroinfiltration

pTRV2-*HhCLA1* was transformed into *Agrobacterium tumefaciens* strain 'GV3101' using the freeze-thawing method (*Höfgen & Willmitzer, 1988*). PCR-confirmed single colonies were then selected and independently inoculated into three mL of Luria-Bertani medium containing 25 mg/L rifampicin and 50 mg/L kanamycin and grown overnight in a shaker at 28 °C. For the VIGS assay, 3-mL cultures of *A. tumefaciens* strain GV3101 independently containing either pTRV1 or pTRV2 was grown overnight in the same culture conditions. These overnight starter cultures were subsequently used to inoculate 50-mL cultures that were grown overnight at 28 °C. *Agrobacterium* cultures were harvested by centrifugation at 4,000× g for 10 min, and the pellets were resuspended in an infiltration buffer (10 mM MES (2- (4- Morpholino) Ethanesulfonic Acid), 10 mM $MgCl_2$ and 200 μM acetosyringone, pH 5.6) at an optical density of 2.0 at 600 nm and incubated at room temperature for 3 h without shaking. *Agrobacterium* cultures containing mixtures of pTRV1 and pTRV2-*HhCLA1* (1: 1 ratio) were infiltrated with 1-mL needleless syringes into the backs of cotyledons of 2-week-old *H. hamabo* seedlings, following a protocol described previously (*Gao et al., 2011*). To determine whether the TRV vector can directly infect *H. hamabo*, a mixture of *Agrobacterium* cultures containing pTRV1 and pTRV2 constructs in a 1: 1 ratio was infiltrated into the backs of cotyledons of eight 2-week-old *H. hamabo* to serve as the mock. Experimental and non-injected control plants were transferred to a growth chamber and maintained under set conditions.

## Quantitative real-time PCR (qPCR)

To determine the relative levels of the endogenous *HhCLA1* transcripts in infected leaves exhibiting visible silencing phenotypes, qPCR was performed using the primer pair q*HhCLA1*-F/q*HhCLA1*-R (Table 1). For the experiments, leaves from plants with significant white streak symptoms were analyzed in comparison with leaves of the mock and control plants after three weeks of agroinfiltration. Four groups of plants with significant white streak symptoms, one control group, and one mock group, in order to analyze the test results more accurately, were further analyzed in this experiment. Each group contained three biological replicates. Total RNA was extracted from these leaves using a Plant RNeasy Mini Kit (Qiagen) and treated with DNase I to remove residual DNA. The first-strand cDNA was synthesized using a SuperScript II reverse transcriptase kit (TaKaRa). The qPCR assays were performed using the SYBR Green PCR Master Mix (Bimake, Houston, TX, USA) and a StepOne™ System (ABI, USA). The transcript level of *18S rRNA* served as the internal controls. All experiments were repeated three times. The relative gene expression level was calculated using the $2^{-\Delta\Delta Ct}$ method (*Gu et al., 2018*; *Liao et al., 2016*).

## Statistical analysis

One-way analysis of variance (ANOVA) and Duncan's multiple range test ($P < 0.05$) were performed using IBM SPSS (Version 21).

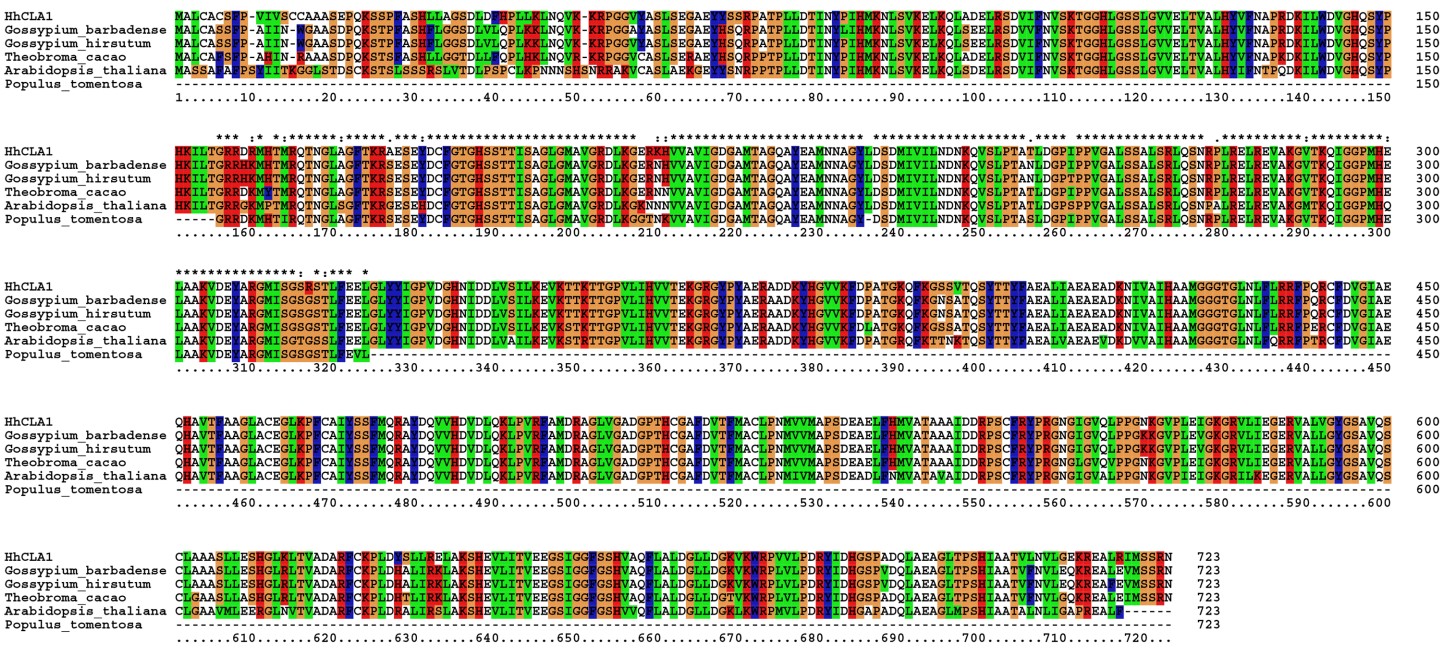

**Figure 1** **Multiple alignment of the HhCLA1 amino acid sequence with sequences from different species using the ClustalW program.** Multiple alignment of protein sequences of the *HhCLA1* gene in *Hibiscus hamabo* Sieb. et Zucc., *Gossypium barbadense* (ABN13970.1), *Gossypium hirsutum* NP_001314056.1), *Theobroma cacao* (EOY06359.1), *Arabidopsis thaliana* (NP_193291.1) and *Populus tomentosa* (AGT02336.1).

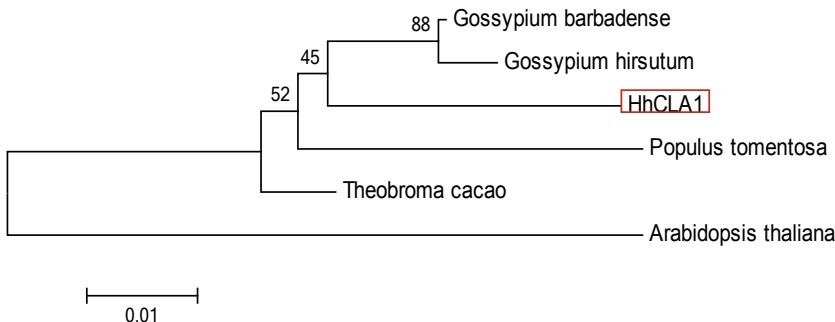

**Figure 2** **Phylogenetic analysis of the protein of HhCLA1.** Phylogenetic analysis of HhCLA1 proteins in different species.

# RESULTS

## Characterization of the *HhCLA1* gene in *H. hamabo*

The amino acid sequence alignment indicated that the HhCLA1 protein showed high homology to known CLA1 proteins from *G. barbadense*, *G. hirsutum* and other species (Fig. 1 and Table S1). The phylogenetic analysis showed that HhCLA1 clustered with *G. barbadense* and *G. hirsutum* in a clade (Fig. 2).

**Table 2**  Efficiency of *HhCLA1* gene silencing in *Hibiscus hamabo* using TRV-VIGS system at three weeks post agroinfiltration.

| Treatment | Number of plants assayed | Silencing efficiency[*] |
|---|---|---|
| pTRV2-*HhCLA1* | 52 | 45/52 (87%) |
| Mock | 8 | 0/8 (0%) |
| Control | 8 | 0/8 (0%) |

Notes.

[*]Silencing efficiency indicates the number of plants showing silencing phenotypes/number of plants treated by TRV-VIGS system.

## Silencing efficiency of the *HhCLA1* gene in *H. hamabo* using the VIGS system

In total, fifty-two *H. hamabo* plants were inoculated with *A. tumefaciens* 'GV3101' harboring pTRV2-*HhCLA1*. Two weeks after agroinfiltration, white streaks began to appear in the emerging leaves of partially agroinoculated plants. At three weeks post infiltration, 87% of the *H. hamabo* plants showed white-streak leaf symptoms similar to the photobleached phenotype (Table 2; Fig. 3A and Fig. S2). At three weeks after agroinfiltration, plants inoculated with pTRV1 and pTRV2 (Mock) showed no obvious differences in leaf morphology compared with the control (Figs. 3B, 3C and Fig. S2). The leaves in Fig. 3D are from plants infiltrated with pTRV2-*HhCLA1* (*CLA1*), empty vector infiltrated plant (Mock) and the control plant (CK) separately. Leaf phenotypic characteristics suggested that the *HhCLA1* gene expression might be suppressed in plants infiltrated with pTRV2-*HhCLA1* compared with mock and CK plants.

## q-PCR analysis of the knockdown levels of *HhCLA1*

The efficiency of gene silencing was analyzed by monitoring expression levels of *HhCLA1* in plants showing white-streak leaf symptoms. Results showed that *HhCLA1* gene expression levels were unchanged in mock-injected plants, while the *HhCLA1* expression levels were 62.6%-76.4% lower in the pTRV2-*HhCLA1* agroinfiltrated plants than in the non-infiltrated plants (control) (Fig. 4 and Table S2 ). The phenotypic characteristics were consistent with the expression characteristics of *HhCLA1*. This clearly indicates that the expression of *HhCLA1* was significantly down-regulated through TRV-VIGS in *H. hamabo*, and TRV-VIGS led to an albino phenotype on leaves.

## DISCUSSION

In this study, we demonstrated for the first time that TRV-VIGS can effectively down-regulate endogenous gene expression levels in the salt-tolerant species *H. hamabo*. The genetic transformation of this species is currently laborious, time-consuming and technically challenging. To resolve these problems, effective and low-cost techniques need to be developed to enable the rapid validation of gene functions. In future studies, stress-responsive genes isolated in *H. hamabo* could be silenced in loss-of-function screens using the TRV-VIGS system.

The *CLA1* gene is involved in chloroplast development and is a useful marker in the TRV-VIGS system (*Mustafa et al., 2016*). In this research, multiple sequence alignments

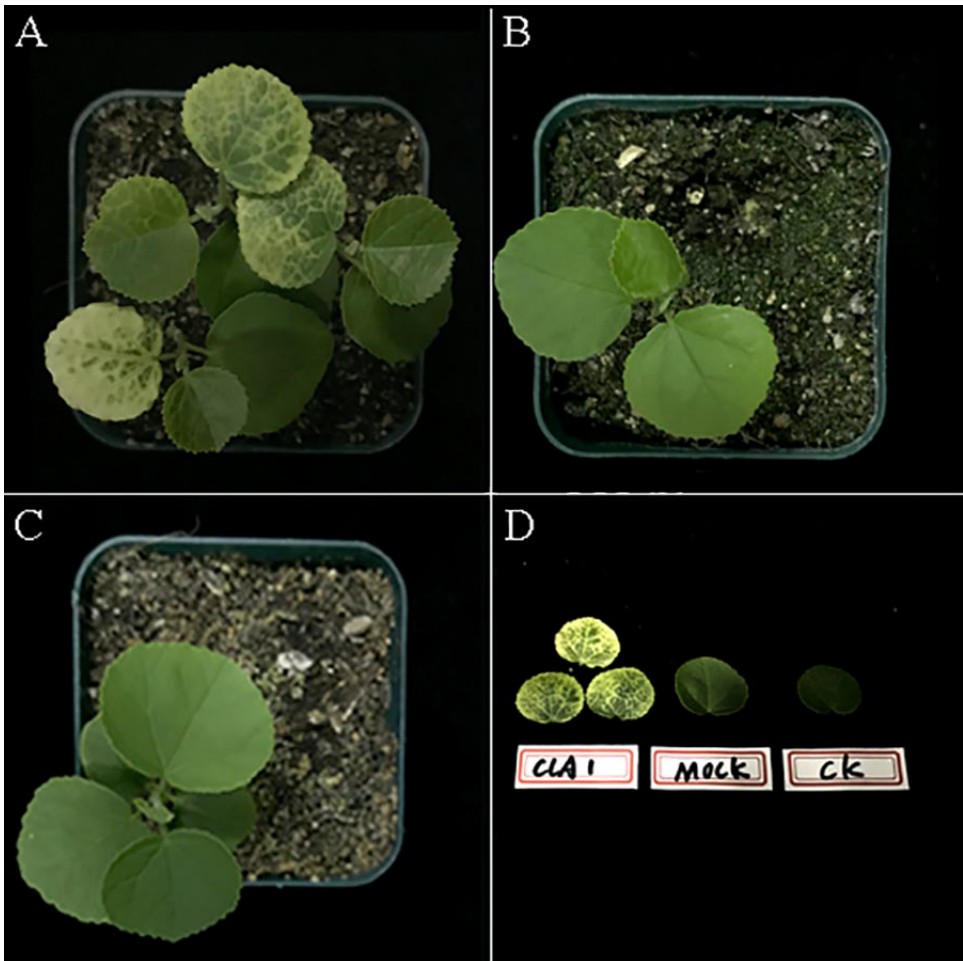

**Figure 3** **TRV-induced *HhCLA1* silencing in *H. hamabo*.** (A) Newly formed leaves of *H. hamabo* plants infiltrated with pTRV2-*HhCLA1* (*CLA1*) showing white-streaked leaf symptoms after three weeks. (B) Empty vector infiltrated plants (Mock) with the normal phenotype. (C) Control plants (CK). (D) The leaf phenotypes of the treatments. The three leaves on the left in Fig. 3D are from plants infiltrated with pTRV2-*HhCLA1* (*CLA1*), the leaf in the middle is from an empty vector infiltrated plant (Mock) and the right one is from a control plant (CK).

indicated that HhCLA1 was similar to CLA1 proteins of other species. Additionally, the phylogenetic analysis indicated that HhCLA1 was highly similar to CLA1 proteins in Malvaceae, including *G. barbadense* and *G. hirsutum*.

The most cost-efficient and effective method of inoculating plants with virus-based vectors is agroinfection (*Grimsley et al., 1986*), but its efficiency varies among plants (*Zhang et al., 2016*). In turf grass, the silencing efficiency of the RTBV-VIGS system in *Cynodon dactylon* was such that 65.8%- 72.5% of the agroinfected plants developed symptoms typical for phytoene desaturase gene silencing, while the silencing efficiency in *Zoysia japonica* was much lower, with only 52.7%–55% of agroinfected plants developing the phenotype (*Zhang et al., 2016*). The ability of the TRV vector to directly infect woody plant species has been tested, and TRV-mediated VIGS was effective in *Vernicia fordii*, weak

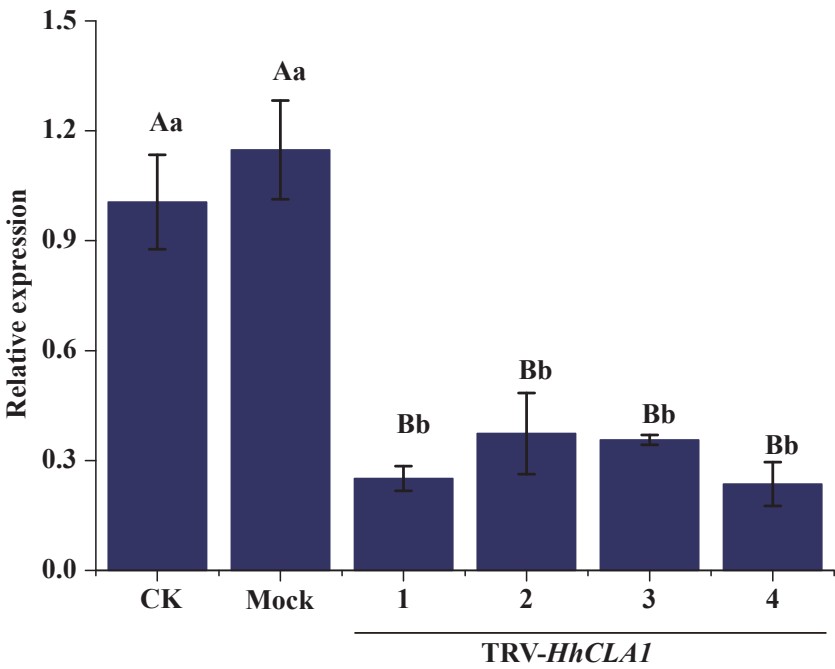

**Figure 4** **Relative expression levels of *HhCLA1* transcripts in control (CK), empty vector-infected (Mock) and pTRV-*HhCLA1*-infected plants (pTRV2-*HhCLA1*).** Error bars represent standard errors, and any two samples with a common capital letter are not significantly different at the $P < 0.01$ level, as with a same small letter are not significantly different at the $P < 0.05$ level.

in *Populus tomentosa* Carr., and ineffective in *Camellia oleifera* (*Jiang et al., 2014*). In this study, the silencing efficiency of the TRV-VIGS system in *H. hamabo* was high, with 87% of agroinfected plants developing a white-streak leaf phenotype. The *HhCLA1* mRNA level was also down-regulated by TRV-VIGS in *H. hamabo*.

# CONCLUSIONS

In conclusion, we demonstrated that TRV-mediated VIGS can effectively silence genes in *H. hamabo*, which adds to the increasing list of wood species for which VIGS-mediated studies can be used. The loss-of-function assay using TRV-mediated VIGS developed in this study provides an alternative tool for functional genes studies of *H. hamabo*.

# ACKNOWLEDGEMENTS

We thank International Science Editing for editing this manuscript.

## Funding

The study was supported by Six Talent Peaks Project of Jiangsu Province (NY-042), Jiangsu Province Three New Forestry Projects (LYSX[2016]53) and the 333 Talents Project of

Jiangsu Province (BRA2017498). The funders had no role in study design, data collection and analysis, decision to publish, or preparation of the manuscript.

## Grant Disclosures

The following grant information was disclosed by the authors:

Six Talent Peaks Project of Jiangsu Province: NY-042.

Jiangsu Province Three New Forestry Projects: LYSX[2016]53.

333 Talents Project of Jiangsu Province: BRA2017498.

## Competing Interests

The authors declare there are no competing interests.

## Author Contributions

- Zhiquan Wang conceived and designed the experiments, performed the experiments, analyzed the data, contributed reagents/materials/analysis tools, prepared figures and/or tables, authored or reviewed drafts of the paper, approved the final draft.
- Xiaoyang Xu conceived and designed the experiments, performed the experiments, analyzed the data, contributed reagents/materials/analysis tools, authored or reviewed drafts of the paper.
- Longjie Ni and Jinbo Guo performed the experiments.
- Chunsun Gu conceived and designed the experiments, performed the experiments, contributed reagents/materials/analysis tools, authored or reviewed drafts of the paper, approved the final draft.

## Data Availability

The HhCLA1 sequence described here is available at GenBank, accession no. MK229167.

## Supplemental Information

Supplemental information for this article can be found online at http://dx.doi.org/10.7717/peerj.7505#supplemental-information.

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
