# Peer review of "Efficient virus-induced gene silencing in Hibiscus hamabo Sieb. et Zucc. using tobacco rattle virus"

_PeerJ, doi:10.7717/peerj.7505_

## Round 0.1 · original submission · Major Revisions

I agree with the reviewers' comments. Please revise your manuscript to address their concerns. Thanks.

Reviewer 1 ·

Basic reporting

No comment

Experimental design

Figure 3B and 3C: The numbers of the mock and the control plants should be more than three. Here, one seedling was only got for each treatment.

Figure 4 and Table S1 show us four individual repetitions of the qPCR results using infected leaves. And the Figure 3A displays only three infected plants. The authors should name each infected plants.

I suggest the authors compare the transcription abundances of HhCLA1 with the chlorophyll contents in each gene silencing plants.

Validity of the findings

This manuscript tried to set up a virus-induced gene silencing system in H. Hamabo. And the transcription abundance of the marker gene (CLA1) in infected leaves was down-regulated. It was a praisable exploration in such an important plant like Hibiscus Hamabo.

Additional comments

On the whole, the research details and writing of the manuscript is some stuffless.

There are several print mistakes in the manuscript.

·

Basic reporting

This manuscript is well writen and the data presented is clear.
The refences about study on Malvaceae in TRV-VIGS technique should be added in Introduction.

Experimental design

experimental design is OK.

Validity of the findings

The findings is important to promote molecular breeding and gene function identification in H. hamabo.

Additional comments

This manuscript reports development of TRV-VIGS technique used a new plant species, Hibiscus hamabo through silencing a marker gene HhCLA1. Authors analyzed HhCLA1 similarities of H. hamabo to different plant species, and succeed in development of HhCLA1-silenced plants through presence of white steak on leaves and increase of HhCLA1 expression level.
This manuscript is well written and the data is clearly presented. The provided information is important to carry out the molecular breeding and gene function identification in H. hamabo.
Some points for authors’ consideration:
1.Line 143-145, the rest of 7 treated plants likely showed subtle white-streak leaf symptoms 2 weeks later, so these treated plants should be monitored about their expression level of CLA1. The experiment can improve the manuscript in quality.
2.The leaf phenotypes shown in Figure 4D are shortage of statement in Results. please check.
3.Line 166, add space to separate "fromother" into "from other"
4.Line 167, studies on VIGS in Malvaceae plants, major different cotton species, should be in more detail discussed for explaining highlight of this research.
5.ref Fowler (2017) is incomplete, please check it.
6.In Figure 4 legends, 1, 2, 3 and 4 individual plants are representatives from 52 treated plants? However, supplemental Table 2 only showed 4 individual silenced plants. Please check. Correct “tandard errors” into “standard errors”

Reviewer 3 ·

Basic reporting

The manuscript entitled “Efficient virus-induced gene silencing in Hibiscus hamabo Sieb. et Zucc. using Tobacco rattle virus” by Zhiquan Wang et al. describes the TRV-mediated gene silencing of Hibiscus plant.
The author demonstrated that the TRV-mediated transient gene silencing is possible in Hibiscus plant and the optimized methodology would be critical for the functional genomics studies of this sequenced plant genome.
Basic reporting is clear and literatures were cited properly.

Experimental design

Experiments design were acceptable except few points need to be addressed as mentioned below.

Validity of the findings

Results are acceptable but need more control to validate the findings, which is mentioned below.

Additional comments

I have many concerns, especially with the figures. Figures arrangement and description require attention. Also controls are missing which I had listed below,

1. In Introduction, author need to elucidate the CLA1 gene in Hibiscus, a comparison of CLA1 gene of Hibiscus with Gossypium spp., Arabidopsis and Vernicia fordii is highly recommended.

2. Although authors had mentioned in materials and method section, I would suggest providing the vector map of the plasmid constructed in figure 3.


3. After infiltration, author observed the appearance of white streak on plant leaves, but no molecular evidence has been provided that this phenotype is a result of the dsRNA which is cleaved into short interfering (si) RNAs of 21 to 25 nucleotides. Any RNA blot or fluorescent/non-radioactive probe to detect these 21-25 nucleotide guide sequences could have been utilized to confirm this working of TRV system and phenotype.

4. Figure 1: The figure is not clear and hard to read, conserved amino acids, motif and domain analysis are also not clear. Define the color code used in figure legend. Mark clearly the conserved residues. I would suggest, for the domain analysis and homology study author should keep only closely related and well characterized 3-5 sequence of CLA1 from different organism. And include Gossypium spp., Arabidopsis and Vernicia fordii as TRV mediated gene silencing has already been performed in this organism.

5. In figure 2 also, include Arabidopsis and Vernicia fordii sequences as well.

6. Result subheadings are poorly written, it should be more conclusive. What do you mean by qPCR??? Are you going to explain qPCR technique??

7. In result section “Characterization of the HhCLA1 gene.” Elaborate in detail about your result of homology and phylogenetic analysis as mentioned above for figure 1.

8. Figure legend is poorly written. Please explain the figures. Remove sequence identity from figure legend 1 and put it in table form.

Overall the manuscript is satisfactory. The findings are worth to report and will be useful in functional genomics of Hibiscus hamabo. Literature is well referenced & relevant to the concept.

---

## Round 0.2 · Major Revisions

As mentioned by Reviewer 1, the manuscript does not meet the publication criteria because of language issues. A lot of edits are needed. I tried to revise your manuscript. But, apparently, it is a daunting task to get it done quickly. Please revise the manuscript thoroughly, and address reviewers' concerns (reviewer 1 and 2). Fluent English is a basic requirement.

Reviewer 1 ·

Basic reporting

In the new version, the manuscript has been made some improvement.

Experimental design

I think the authors answer my questions below clear.
1. Figure 3B and 3C: The numbers of the mock and the control plants should be more than three. Here, one seedling was only got for each treatment.
2. Figure 4 and Table S1 show us four individual repetitions of the qPCR results using infected leaves. And the Figure 3A displays only three infected plants. The authors should name each infected plants.
3. I suggest the authors compare the transcription abundances of HhCLA1 with the chlorophyll contents in each gene silencing plants.

Validity of the findings

It is a worthy effort for establish of a gene silencing system in Hibiscus Hamabo.

Additional comments

The manuscript needs much more attentions on English writing.

·

Basic reporting

as shown in general comments to authors

Experimental design

as shown in general comments to authors

Validity of the findings

as shown in general comments to authors

Additional comments

The revised manuscript has been obviously improved in quality. The most questions have been explained. I have not other comments for this revised manuscript excluding some minor suggestions.
1. “Gossypium” throughout the manuscript should be omitted to “G.” excluding the first presence.
2. Line 205, “functional genomics” is unfit for VIGS tool because it is used in transcriptional level.
3. In figue 4, “Aa” and “Bb” are unclear. Please check it.

Reviewer 3 ·

Basic reporting

The manuscript entitled “Efficient virus-induced gene silencing in Hibiscus hamabo Sieb. et Zucc. using Tobacco rattle virus” by Zhiquan Wang et al. describes the TRV-mediated gene silencing of Hibiscus plant.
Manuscript has improved after revision and is acceptable for publication.

Experimental design

Satisfactory

Validity of the findings

Satisfactory

Additional comments

I agree with the authors about their response for my comment especially number 3. Answer to this question may qualify for the next publication.

---

## Round 0.3 · Major Revisions

The manuscript, especially the English language, still needs to be significantly improved. Please consider my comments, and revise your manuscript (see the attachment). Native speakers of English may help you improve the manuscript.

---

## Round 0.4 · Minor Revisions

The manuscript was significantly improved. However, minor revisions are still needed. I made some comments and revisions on the attached manuscript. In addition, I also suggest that you further revise the legends of Fig. 1 and Fig.2 etc:

Fig. 1: Multiple alignment of protein sequences of the HhCLA1 gene in Hibiscus hamabo Sieb. et Zucc., Gossypium barbadense (ABN13970.1), Gossypium hirsutum (NP_001314056.1), Theobroma cacao (EOY06359.1), Arabidopsis thaliana (NP_193291.1) and Populus tomentosa (AGT02336.1).
Fig. 2 Phylogenetic analysis of HhCLA1 proteins in different species
Table 1 and Table 2: the RV-VIGS system

Please consider my comments and suggestions, and revise your manuscript. Thanks.

---

## Round 0.5 · accepted · Accept

The manuscript has been significantly improved, and is acceptable. Many thanks for your revisions and resubmission.